

# Myriad applications of bacteriophages beyond phage therapy

Juan Carlos García-Cruz[1,*], Daniel Huelgas-Méndez[1,*],
Jorge Santiago Jiménez-Zúñiga[1], Xareni Rebollar-Juárez[1],
Mariel Hernández-Garnica[1], Ana María Fernández-Presas[1],
Fohad Mabood Husain[2], Rawaf Alenazy[3], Mohammed Alqasmi[3],
Thamer Albalawi[4], Pravej Alam[4] and Rodolfo García-Contreras[1]

[1] Faculty of Medicine Department of Microbiology and Parasitology, Universidad Nacional de México, Mexico City, Mexico
[2] Department of Food Science and Nutrition, King Saud University, Riyadh, Saudi Arabia
[3] Department of Medical Laboratory, College of Applied Medical Sciences-Shaqra, Shaqra University, Shaqra, Saudi Arabia
[4] Department of Biology, Prince Sattam bin Abdulaziz University, Alkharj, Saudi Arabia
* These authors contributed equally to this work.

## ABSTRACT

Bacteriophages are the most abundant biological entity on the planet, having pivotal roles in bacterial ecology, animal and plant health, and in the biogeochemical cycles. Although, in principle, phages are simple entities that replicate at the expense of their bacterial hosts, due the importance of bacteria in all aspects of nature, they have the potential to influence and modify diverse processes, either in subtle or profound ways. Traditionally, the main application of bacteriophages is phage therapy, which is their utilization to combat and help to clear bacterial infections, from enteric diseases, to skin infections, chronic infections, sepsis, *etc*. Nevertheless, phages can also be potentially used for several other tasks, including food preservation, disinfection of surfaces, treatment of several dysbioses, and modulation of microbiomes. Phages may also be used as tools for the treatment of non-bacterial infections and pest control in agriculture; moreover, they can be used to decrease bacterial virulence and antibiotic resistance and even to combat global warming. In this review manuscript we discuss these possible applications and promote their implementation.

## INTRODUCTION

Bacteriophages are the most abundant biological entity on the planet, likely surpassing bacteria by a factor of 10 (*Mushegian, 2020*). They play important roles in global ecology, biogeochemical cycles, and animal and plant health (*Batinovic et al., 2019*). Bacteriophages were discovered independently by Frederick Twort and Felix d'Herelle, and d'Herelle hypothesized that they were viruses as well as coined the term bacteriophage. d'Herelle also pioneered phage therapy, which is the utilization of phages to treat infections. From its beginning, phage therapy was very promising and useful for the treatment of dysentery, cholerae and other diseases, and its study and implementation continued to be carried out

Corresponding author
Rodolfo García-Contreras,
rgarc@bq.unam.mx

in some countries in Eastern Europe like Georgia and Poland. However, due the discovery of antibiotics, the use of phage therapy in the western world was scarce (*Summers, 2012*). Nevertheless, due the increasing prevalence in bacterial resistance towards antibiotics, recently the use of bacteriophages in occidental medicine has been revived, especially as the last resource to treat recalcitrant infections caused by multi- and pan-drug-resistant bacteria. Hence, the number of successful cases in which phage therapy was able to rescue untreatable patients is increasing (*Schooley et al., 2017*; *LaVergne et al., 2018*; *Law et al., 2019*; *Johri et al., 2021*). Recently, Strathdee and coworkers published a comprehensive review on phage therapy explaining phage biology, the ideal characteristics of phages that make them suitable for their utilization in phage therapy, phage engineering and current phage mainstream applications (*Strathdee et al., 2023*). Remarkably, bacteriophages have a great versatility and potential as useful tools in several fields beyond classic phage therapy, including cattle raising, agriculture, pest control, microbiome modulation, and disinfection. In this review, we discuss several possible bacteriophage potential and robust applications beyond classical phage therapy and hope it will stimulate research for these applications and lead to additional uses. This review is intended for microbiologists, biotechnologists and to a general audience interested in biotechnological applications of bacteriophages.

## SURVEY METHODOLOGY

Suitable literature was compiled using the main online scientific databases, including PubMed, Science Direct (http://sciencedirect.com), Web of Science, Scopus, and Google Scholar. Several keywords, such as bacteriophage, applications, antibiotic resistance, virulence, quorum sensing, biofilm, disinfectants, food preservatives, probiotics, microbiome, dysbiosis, CRISPR-Cas, methane, global warming, *etc.*, were used, together with Boolean operators such as "AND" and "OR".

### Bacteriophages as preservatives, disinfectants, and biofilm inhibitors

Bagged products, such as salads, greens and mushrooms, represent environments suitable for bacterial growth, and contamination has resulted in several outbreaks, like 2022's *Listeria monocytogenes* outbreaks in the U.S. *Bacillus cereus*, *Campylobacter jejuni*, *Clostridium perfringens*, *Escherichia coli*, *Salmonella spp.*, *Shigella spp.* and *Staphylococcus aureus* are some of the most common pathogens that contaminate and spoil food, nevertheless pathogens like *Borrelia spp.* and *Yersinia enterocolitica* are also of concern. Phage utilization after the harvest of vegetables or before food packaging should be further evaluated, and there are several commercial products like Intralytix, Micros Food Safety, FINK TEC GmbH, Passport Food Safety Solutions and Phagelux, which have gained approval of the Food and Drug Administration (FDA) (*Arienzo et al., 2020*; *Endersen & Coffey, 2020*). Also, phage utilization in the food industry may be useful for preventing contamination by *Cronobacter sakazakii*, in baby formula and *Clostridium botulinum* in honey. Both pathogens affect children and present misleading clinical presentations.

Furthermore, phage preparations may also be useful as self-replicating disinfectants for decontaminating or cleaning surfaces, such as in hospitals. They are particularly effective

at killing bacteria within biofilms since biofilms are often resistant to chemical disinfectants. To date, it has been shown that for some bacteria, such as *Pseudomonas aeruginosa*, chemical disinfectants like sodium hypochlorite and benzalkonium chloride have synergistic effects with some phages, enhancing biofilm elimination on contaminated surfaces. Similarly, phages that can eliminate biofilms of other important bacterial pathogens such as *S. aureus, E. coli, Acinetobacter baumannii, Klebsiella pneumoniae, etc.*, have been reported (*Song et al., 2021*; *Liu et al., 2022*).

## Bacteriophages as tools for decreasing antibiotic resistance and bacterial virulence

The ability of bacteria to readily develop resistance against phages is a double-edged sword for bacterial pathogens, since one of the most common bacterial mechanisms of developing phage resistance is to mutate the phages receptors, frequently diminishing or avoiding its expression; however, some of the phages preferred receptors are structures related to virulence, such as flagella, type IV pilli, and capsular polysaccharide and some others such as efflux pumps and lipopolysaccharide (LPS) are involved in bacterial antibiotic resistance. Hence, bacteria that become resistant to phage sometimes also decrease their virulence and/or their antimicrobial resistance. Several authors had already demonstrated this property for important bacterial pathogens such as *P. aeruginosa* and *A. baumannii* (*León & Bastías, 2015*; *Chan et al., 2016*; *Gordillo Altamirano et al., 2021*). Moreover, important processes leading to the exchange of genetic material in bacteria such as conjugation, which promotes the spread of bacterial resistance, can also be inhibited by some filamentous phages for example by the occlusion of conjugative pilus by phage particles (*Lin et al., 2011*). Hence, perhaps this property could be used to decrease bacterial virulence bacteria or to re-sensitize pathogenic bacteria against antibiotics in hospitals, farms, aquaculture, and wastewater treatment plants and even in the environment. Their implementation may be first done in small experimental settings to evaluate its performance and gradually could be scaled. Furthermore, phages may also be used to inhibit quorum sensing (QS), which is a mechanism of cell signaling that allows the regulation of gene expression in response to changes in cell-population density. Many virulence genes are upregulated through this mechanism, therefore the inhibition or interruption of the QS signaling pathway is a desired mechanism of action of antivirulence compounds (*Asfour, 2018*). For example, (Aqs1) located in the genome of the *Pseudomonas aeruginosa* phage DMS3 contains two domains: one that inhibits the protein LasR, the master regulator of QS in this bacterium, and another that inhibits the activity of the type IV pilus assembly ATPase protein (PilB). Remarkably, this protein requires 69 residues to decrease virulence (*Shah et al., 2021*). Hence, phages with these kinds of negative regulators of QS, may be also useful for reducing bacterial virulence.

## Phages to treat dysbiosis and modulate microbiomes

Animal microbiomes, including humans, are dominated by bacterial species and microbiome alterations have been linked to a plethora of diseases including obesity, type II diabetes, autoimmune diseases, cancer, autism, depression, periodontal disease (*Wang*

*et al., 2017*). Phages could be an ideal tool for the treatment of several dysbioses, helping to reach an adequate balance of the microbiome, particularly if the imbalance is caused by one or a few species that promote disease when they overgrow. For example phages could be implemented to control the populations of *Helicobacter pylori*, the main agent causing gastric and peptic ulcers, especially since current triple and quadruple therapies with antibiotics aim to eliminate it, but the evidence of a protective role of this bacterium against a variety of diseases such as asthma, allergies, and esophageal cancer is growing (*Shahabi et al., 2008*; *Zuo et al., 2021*). Hence, perhaps modulation of their population size, rather than complete elimination, could be a better option for enhancing human health. Unfortunately, there are only few examples of characterized *H. pylori* bacteriophages and most of them are lysogenic (*Muñoz et al., 2020*; *Khosravi et al., 2021*). Therefore, further research is required, which will allow the identification of lytic phages suitable for controlling *H. pylori* populations in patients. Another possible bacterial target for phages is *Clostridium difficile*, which causes recurrent diarrhea usually in elderly people after an antibiotic course due to overgrowth of this bacterium that survives the antibiotics since it produces spores that are metabolically inactive and hence insensitive to antibiotics. Currently, the most successful treatment for those patients is a fecal transplant from healthy donors (*Juul et al., 2018*); however, phages able to kill *C. difficile* may be another possibility and these have the advantage that they may be used prophylactically in patients with probabilities of develop recurrent diarrhea before or during the antibiotic course (*Hargreaves & Clokie, 2014*). Furthermore, other dysbioses such as acne, the most common skin infection, is produced by an overgrowth of the bacterium *Cutibacterium acnes* that triggers inflammatory responses; to date several phages active against *C. acnes* have been isolated and studied, finding that usually they are able to infect several *C. acnes* strains which is a good quality for their possible application (*Golembo et al., 2022*). Another dysbiosis that potentially could be treated with bacteriophages is vaginosis, which is caused by the overgrowth of *Gardnerella vaginalis*, a natural inhabitant of the female genital tract that is normally non harmful unless it overgrows. Vaginosis is usually treated with the antibiotics clindamycin or metronidazole but relapses frequently; hence, phage therapy could be another alternative to combat this common dysbiosis (*Bordigoni, Bouchard & Desnues, 2021*). Also, in a more generic way, phages can be added to prebiotic and probiotic formulations to enhance their health promoting effects by reducing the growth of potentially harmful bacteria (*Kiani et al., 2020*) or by decreasing bacterial competitors of health-promoting species. More complex multifactorial dysbioses, such as those causing periodontal disease, dental bacterial plaque and caries, are also susceptible to be treated using bacteriophages. For instance, several bacterial pathogens such as *Streptococcus mutans*, which is likely the main causative of dental caries, and *Aggregatibacter actinomycetemcomitans* which is the bacteria most frequently related to rapidly progressive periodontitis, are strong candidates for the application of phages as they are important periodontal pathogens and since lytic phages able to destroy these bacteria either in a free living state or in biofilms have been described (*Castillo-Ruiz et al., 2011*; *Dalmasso et al., 2015*). Moreover, health-damaging conditions such as obesity may be also
combated by administrating phages targeting bacteria associated with a promotion of weight gain, fat accumulation, *etc.*

## Bacteriophages to combat nonbacterial diseases

Some degenerative diseases like Parkinson or Alzheimer are characterized by the accumulation of misfolded protein aggregates which possess a canonical amyloid fold. The current therapies for these diseases are based on targeting specific protein aggregation pathways. In an interesting work, the bacteriophage M13 capsid protein was shown to have a motif that interacts with the canonical amyloid fold; therefore, it has been used to reverse the formation of plaques derived from amyloid-like structures in the brain. This protein represents a novel approach to combat neurodegenerative diseases (*Krishnan et al., 2014*). There is a famous Arab quote that says: "*The enemy of my enemy is my friend*". In phage-based technologies, this phrase is not only appropriate for the context of combating bacterial infections using classical phage therapy but also for viral infections in animals and plants. Most of these properties of phages against viral infections are of immunomodulatory nature but they also can be used as vehicles for gene delivery, for diagnosis of diseases and for discovery and development of new antivirals, as well as vaccine development. Some antiviral responses that are triggered by the interaction between eukaryotic cells and bacteriophages indicate that phages have the potential to be utilized as adjunct therapy of viral infections like those caused by SARS-COV-2 and Epstein-Barr virus (EBV). Moreover, a secondary benefit of this application of phages would be that it could aim to prevent or combat secondary bacterial infections that often appear and kill coronavirus disease of 2019 (COVID) patients (*Gorski et al., 2020*; *Shahin et al., 2022*). Tumor necrosis factor (TNF) levels of COVID-19 patients are high so anti-TNF therapies are recommended because the elevated production of this anti-inflammatory cytokine can increase the risk of viral replication and bacterial infection (*Feldmann et al., 2020*). Since some phages that infect *A. baumannii* such as φkm18P and Bφ-R2096 downregulate TNF-α concentration in lungs and serum of mice with acute pneumonia, they may be used in human patients for decreasing TNF-α level upon viral infections. Also, it has been hypothesized that the integrin KGD motifs (Lys-Gly-Asp) located on the capsids of some phages such as T4 could compete with SARS-CoV-2 and other coronaviruses for binding their receptors, since the exposed loop of the protein S of SARS-CoV-2 has the same motif (*Gorski et al., 2020*). Furthermore, this motif is also relevant for the infection of the EBV to B and epithelial cells, so phage T4 can also potentially interfere in the adsorption and internalization of EBV (*Górski et al., 2018*). As it was mentioned above, bacteriophages can be used as vehicles for gene delivery. In 2019, Przystal and coworkers used a hybrid vector composed of the phage M13 with a single stranded adeno-associated transgene cassette inserted in an intergenomic region of the single stranded phage genome to treat glioblastoma multiforme (GBM) in mice. The vector bears the RGD4C ligand in the pIII minor coat protein on the capsid of M13 which binds the αvβ3 integrin receptor present in the tumor cells. Then, the hybrid vector is internalized and the recombinant genome is delivered into the nucleus to generate the expression of a therapeutic gene called HSVtk which induces cell death by apoptosis.

Moreover, the expression of HSVtk was under the control of a promoter induced by the administration of TMZ. The approach was efficient and harmless (*Przystal et al., 2019*).

For viral therapeutics, a similar approach can be applied in which phage based vectors can be engineered to carry and deliver CRISPR-Cas systems as therapeutic genes and target an infected eukaryotic cell editing its genome, changing either host or viral genes. This approach could be optimal for those viruses that can integrate into its eukaryotic host genome like human immunodeficiency virus, papillomavirus and herpes simplex viruses (*Chen et al., 2018*; *Chen, Yu & Guo, 2018*). The Clustered Regularly Interspaced Short Palindromic Repeats (CRISPR)-associated (Cas) system includes the helicase called the Cas protein which can bind to RNA transcribed from palindromic repeats of DNA. The transcript of palindromic repeats of DNA is known as the transactivating CRISPR RNA (tracrRNA) while the spacers transcript is called CRISPR RNA (crRNA). The tracer RNA and crRNA can be linked to form a single guide RNA (sgRNA) that can direct Cas9 to induce a double stranded DNA break in the protospacer adjacent motifs (PAMs) region of the DNA target (*Chen, Yu & Guo, 2018*). Despite that there are two effective vaccines against human papilloma virus (HPV), it remains as the causal agent of the second most common cancer worldwide. Moreover, its double stranded DNA (dsDNA) may integrate into the host genome or remain as an episome so it could be also edited with CRISPR-Cas. There are two main target genes of HPV that can be treated with this strategy, the oncogenes E6 and E7, which can induce the degradation of tumor-suppressive protein (p53), and retinoblastoma (Rb), thereby blocking the cell cycle (*Kennedy et al., 2014*). *Kennedy et al. (2014)* proved that the expression of a bacterial Cas9 RNA-guided endonuclease, together with two pairs of single guide RNAs (sgRNAs) specific for E6 and E7 of HPV-18 and HPV-16, respectively, induce cleavage of the HPV genome and eventual cell cycle arrest and cell death in transformed SiHA and HeLa cervical carcinoma cell lines. Although they used a lentiviral vector, it is also possible to use engineered phage-based vectors to improve the efficiency of transduction and this procedure can be escalated to an *in vivo* infection model. Some of the advantages of using engineered phages as a vehicle of CRISPR-cas delivery into mammalian cells are their specific cell tropism, in contrast to eukaryotic virus vectors which are more prone to have interactions with more than one type of eukaryotic cell and consequently more chances to develop side effects, making phage-based vectors safer for clinical use (*Bakhshinejad & Sadeghizadeh, 2014*). Nonetheless, the interactions between phages and the immune system should be investigated further in order to make this strategy safer and more effective. Furthermore, the relatively simple genetics of the phages make them easier to manipulate by widespread using tools like phage display and also phages can package a large sized vector-genomes in contrast to adeno-associated virus vectors (*Zhou, Suwan & Hajitou, 2020*; *Fage, Lemire & Moineau, 2021*).

The challenges for using phages as delivery methods for CRISPR-Cas involve the manufacture of the engineered phages, likely requiring utilization of strict and expensive standards for producing and purifying them.

## Phages as a tool for pest control

The utilization of bacteriophages to combat bacterial diseases in agriculture and farming is a growing practice showing promising results as a safe and efficient alternative to the use of antibiotics. However, beside the straight forward utilization of phages for killing bacteria that cause disease to commercially-important cultures, we propose that they can be used to debilitate and perhaps even kill some insect pests that rely in their microbiome to perform important physiological functions such as bacteria of the genera *Buchnera* that are obligate endosymbionts of aphids and contribute to the production of tryptophan (a key factor to maintain nutritional balance) and for the development and reproduction of their hosts (*Shigenobu & Wilson, 2011*; *Zhang et al., 2021*). Similarly, bacterial symbionts are important for the physiology of other pests such as thermites, flies, mosquitoes, and roaches. So far, an interesting work demonstrated that phages can be used to remove up to 90% of *P. aeruginosa* in the gut of *Musca domestica* and that the removal of this species (that provides the flies defense against fungal infections) leaded to broad changes in gut microbiome composition affecting the normal development of the flies (*Zhang et al., 2021*).

Mosquitoes are the most important vectors involved in the transmission of multiple animal and human pathogens, for that reason several strategies affecting the development of these insects have been investigated. The microbiota of adult mosquitoes is mainly derived from the larval microbiota, which in turn depends on the microbiota of their water habitat; alterations in the microbiota of mosquito larvae have a huge impact on mosquito biology for that reason this approach can potentially be exploited to alter the mosquitoes' life-history traits. Recently it was shown that the microbiota of *Anopheles* larvae could be modulated by using bacteriophages in the larvae water (*Dimopoulos, 2022*). In that work gnotobiotic *Anopheles* larvae with microbiota composed of *Enterobacter*, *Pseudomonas*, *Serratia*, *Elizabethkingia*, and *Asaia* showed a survival and larval development reduction with the addition of phages into the larvae water targeting *Enterobacter* and *Pseudomonas* respectively, and a synergistic behavior was shown when both phages were added simultaneously (*Dimopoulos, 2022*).

## Phages for combating global warming

Methane is the second more abundant greenhouse gas, just behind $CO_2$ and a significant amount of it is produced by methanogenic archaea in the microbiome of cattle raised for human consumption. Since archaea also are infected by viruses (equivalent to bacteriophages), they can be used to decrease their population in cattle microbiome and decrease methane emissions produced by livestock (*Boadi et al., 2004*). Alternatively, phages that target eubacteria that have synergies with methanogenic archaea may be also targeted since the decrease of their populations will also decrease that of the methanogens and hence mitigate methane production. Similarly, $CO_2$ emissions and trapping may be modulated using bacteriophages to control bacterial populations. Figure 1 summarizes the main phage applications discussed in this manuscript.

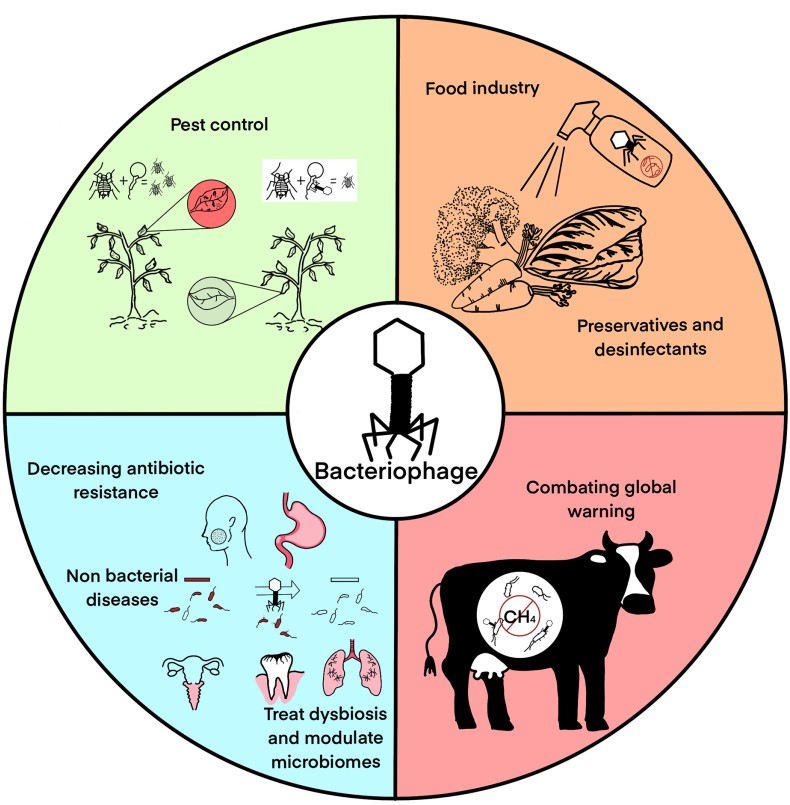

**Figure 1 Main bacteriophage applications: pest control, food industry, medical applications excluding classical phage therapy, and global warming mitigation.**

## Possible limitation and drawbacks

Currently most of the applications for phage therapy described here are still in their early stages, and much more basic research is needed. For example, research is needed to expand the repertoire of lytic phages against *H. pylori*, *C. difficile* and *G. vaginalis*, since to date only a few phages able to infect them have been described and some of them are lysogenic rather than lytic (*Bordigoni, Bouchard & Desnues, 2021*), making it necessary to identify more suitable phages and/or engineer the lysogenic ones to turn them in to lytic. On the other hand, some illnesses originated from dysbioses such as periodontal disease, are complex since several different bacterial species are implicated acting in a specific spatiotemporal order; hence, the design of phage treatments for the prevention and cure of these diseases needs to take these characteristics into account. Furthermore, phage mediated lysis of bacteria sometimes releases toxic bacterial compounds like LPS and other toxins that can damage the host and promote inflammation. Moreover, the high prevalence of phages in the environment is a risk of contamination in the dairy industry, potentially causing reduction or loss of the yield in several dairy products such as yogurt, kefir, and cheese. In addition, the contamination of these products with antibiotic resistance genes transferred by phages may massively disseminate antibiotic resistance (*Blanco-Picazo et al., 2022*); nevertheless, one way to prevent this scenario is to use a

temperate phage to lysogenize industrial strains to exclude environmental phages. Therefore, this approach would provide immunity against phages, mitigating the risk of undesirable phage infections (*da Silva Duarte et al., 2019*). Moreover, despite the myriad of applications bacteriophages have, extensive further research is needed in order to increase our understanding of phage biology and of their ecological relationship with bacteria. For instance, current applied phages are selected avoiding those that carry potential genes encoding toxins and other virulence factors, antibiotic resistance genes, *etc.*; however, around 70% of phage genes have unknown functions hence it is not possible to rule out that some of the used phages have potentially dangerous genes (*Mcnair et al., 2019*). Moreover, ecology is complex and perhaps the utilization of phages for treating dysbioses could have unexpected consequences. For example, for *P. aeruginosa*, some phages can preferentially eliminate mutants defective in quorum sensing (QS) that are impaired in the production of many virulence factors and potentially could increase the fraction of QS proficient bacteria selecting therefore more harmful bacteria (*Saucedo-Mora et al., 2017*; *Cazares, García-Contreras & Pérez-Velázquez, 2020*). Beyond that, the escalation of phage preparations, and the financial and technical challenges to make them commercially available for any application should be considered. Finally, if the utilization of phages becomes widespread, society and industry should avoid committing the same mistakes that lead us to the current antibiotic crisis (*Amábile-Cuevas, 2022*).

## CONCLUSIONS

Although the mainstream bacteriophage application is their use for treating bacterial infections phage therapy, possible alternative bacteriophage applications are numerous and include ones in the industrial, medical, veterinary and health fields. Critically, their possible uses are only limited by the human imagination; hence, in the following decades phages may become fundamental contributors for solving some of the more critical global concerns for humanity, including (1) reducing the antibiotic crisis, not only curing patients infected with bacteria insensitive to antibiotics but contributing to the decrease in the prevalence of multi and pan drug resistant bacteria in hospitals (if they are used as disinfectants) and in the environment (if regular antibiotics used in livestock raising and in agriculture are replaced by phages). Additionally, even if some bacteria turn resistant against the used phages, new phages can be selected and obsolete phages may be engineered or evolved to regain the ability to infect the hosts. Furthermore, sometimes bacteria that become phage resistant suffer a decrease in their antibiotic resistance ability; therefore, this property may also contribute to decrease the overall antibiotic bacterial resistance worldwide. (2) Increasing food production, by their application for combating bacterial pathogens that attack crops and perhaps by their use to debilitate insects, nematodes, and other eukaryotic pests that depend on their bacterial microbiota to perform important metabolic traits. (3) Decreasing obesity if they are used to promote the reduction of obesity promoting bacterial species in the human gut microbiota or by enhancing the proliferation of lean promoting species and (4) reducing global warming, if they are used to reduce the populations of methanogenic archaea or methanogenic promoting bacteria in cattle.

Furthermore, we think that many novel ways to utilize bacteriophages or their components will constantly appear, increasing even more the potential benefits of the utilization of these viruses.

On the other hand, several considerations such as possible not desired effects of their applications and challenges such as massive production of phages and their thoughtful purification need to be contemplated in order to maximize the benefits mankind can obtain from these versatile and amazing organisms.

## ACKNOWLEDGEMENTS

We are grateful with Professor Dr. Thomas K. Wood, from Penn State University for proofreading the manuscript.

### Funding

This study is supported *via* funding from Prince Sattam bin Abdulaziz University project number (PSAU/2023/R/1444) and by DGAPA-UNAM grant number PAPIIT-IN200121. The funders had no role in study design, data collection and analysis, decision to publish, or preparation of the manuscript.

### Grant Disclosures

The following grant information was disclosed by the authors:
Prince Sattam bin Abdulaziz University: PSAU/2023/R/1444.
DGAPA-UNAM: PAPIIT-IN200121.

### Competing Interests

Rodolfo García-Contreras is an Academic Editor for PeerJ.

### Author Contributions

- Juan Carlos García-Cruz conceived and designed the experiments, performed the experiments, analyzed the data, authored or reviewed drafts of the article, and approved the final draft.
- Daniel Huelgas-Méndez conceived and designed the experiments, performed the experiments, analyzed the data, authored or reviewed drafts of the article, and approved the final draft.
- Jorge Santiago Jiménez-Zúñiga conceived and designed the experiments, performed the experiments, analyzed the data, authored or reviewed drafts of the article, and approved the final draft.
- Xareni Rebollar-Juárez conceived and designed the experiments, performed the experiments, analyzed the data, authored or reviewed drafts of the article, and approved the final draft.
- Mariel Hernández-Garnica conceived and designed the experiments, performed the experiments, analyzed the data, prepared figures and/or tables, authored or reviewed drafts of the article, and approved the final draft.

- Ana María Fernández-Presas conceived and designed the experiments, performed the experiments, analyzed the data, authored or reviewed drafts of the article, and approved the final draft.
- Fohad Mabood Husain conceived and designed the experiments, performed the experiments, analyzed the data, authored or reviewed drafts of the article, and approved the final draft.
- Rawaf Alenazy conceived and designed the experiments, performed the experiments, analyzed the data, authored or reviewed drafts of the article, and approved the final draft.
- Mohammed Alqasmi conceived and designed the experiments, performed the experiments, analyzed the data, authored or reviewed drafts of the article, and approved the final draft.
- Thamer Albalawi conceived and designed the experiments, performed the experiments, analyzed the data, authored or reviewed drafts of the article, and approved the final draft.
- Pravej Alam conceived and designed the experiments, performed the experiments, analyzed the data, authored or reviewed drafts of the article, and approved the final draft.
- Rodolfo García-Contreras conceived and designed the experiments, performed the experiments, analyzed the data, prepared figures and/or tables, authored or reviewed drafts of the article, and approved the final draft.

## Data Availability

This is a literature review.

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
