# Peer review of "Myriad applications of bacteriophages beyond phage therapy"

_PeerJ, doi:10.7717/peerj.15272_

## Round 0.1 · original submission · Major Revisions

Dear authors,
Two experts on the field have addressed the content and writing of this manuscript. Both experts agree that the manuscript is relevant to the field, is unbiased and the content is good. However, both reviewer 2 and I think that the manuscript requires an extensive revision of the grammar and writing, please make sure to revise the punctuation and the spelling.

Thank you for choosing PeerJ for submitting this excellent review.

Best regards,
Bernardo

·

Basic reporting

The document submitted, is a clear and substantial group of examples about alternative uses of Bacteriophages. This text gathers ideas for the development of these exciting research lines, very well described and explained.
Concerning to the text I have minor corrections in the next appointed lines:

26. most abundant biological entities

101. The term "living disinfectants" implies that bacteriophages are alive and although they are capable of close interaction with bacteria, they lack fundamental characteristics of the living organisms: metabolism and self repair. Instead of "living disinfectants", I suggest alternative organic disinfectants.

114.diminishing or avoiding its expression

125. bacterial virulence or sensitize them

162. this bacterium

224. Explain a little bit further, how can phages interact with eukaryote cells??

272. Buchnera in italic font.

273. contribute to the production.

Experimental design

I agree this reviewing text is within the aims and scope of the journal of Life and Environment.
The ideas discussed in this review text, span from microbiology to ecology.
The text is based on a planed survey and the results reflect the contemporary and resent knowledge of the field.

Validity of the findings

Argumentation of each section is clearly sustained by pertinent references and the redaction is intended to explain the different ideas about the use of phages, as well as pros and cons of these uses. Is not only the state of the art, is a guide.

Additional comments

Clever and clear subject covering in each section.

Reviewer 2 ·

Basic reporting

The authors present an interesting review of possible uses of bacteriophages (phages) beyond typical phage therapy. Although recent reviews exist on this field, most of them focus on the clinical or medical aspect. The present review is logically organized with appropriate sections and subsections, but the English grammatical structure, punctuation, and overall writing style are not professional enough, requiring thorough editing before being suitable for publication. Some things to consider include, but are not limited to, grammatical errors (e.g., wrong conjugations, incorrect use of prepositions, etc.), incorrect punctuation in long paragraphs, which makes them hard to follow, and the lack of definition of many abbreviations on first use (e.g., FDA, LPS, ESKAPE, etc.). Some abbreviations are incorrectly spelled (e.g., CRIPSR instead of CRISPR). The following two paragraphs are examples of most of these grammatical problems, but the entire document must be thoroughly revised for clarity and professional English usage.

Lines 76-80
ORIGINAL.
Suitable literature was compilated using the main online scientific databases: PubMed, Science Direct (http://sciencedirect.com), Web of Science, Scopus, and Google Scholar. Using several keywords including: bacteriophage, applications, antibiotic resistance, virulence, quorum sensing, biofilm, disinfectants, food preservatives, probiotics, microbiome, dysbiosis, CRISPR-Cas, methane, global warming, etc. together with Boolean operators such as ``AND'' and ``OR''.

REVISED.
Suitable literature was compiled using the main online scientific databases, including PubMed, Science Direct (http://sciencedirect.com), Web of Science, Scopus, and Google Scholar. Several keywords, such as bacteriophage, applications, antibiotic resistance, virulence, quorum sensing, biofilm, disinfectants, food preservatives, probiotics, microbiome, dysbiosis, CRISPR-Cas, methane, global warming, etc., were used, together with Boolean operators such as "AND" and "OR".

Lines 101-108
ORIGINAL.
Furthermore, phage preparations may also be useful as living disinfectants in order to decontaminate or to clean surfaces, for example in hospitals, being especially useful for killing bacteria within biofilm since biofilms are often resistant to chemical disinfectants, to date it has been shown that for some bacteria such as Pseudomonas aeruginosa chemical disinfectants such as sodium hypochlorite and benzalkonium chloride have synergistic effects with some phages, enhancing biofilm elimination in contaminated surfaces, similarly phages able to eliminate biofilms of other important bacterial pathogens such as S. aureus, Escherichia coli, A. baumannii, K. pneumoniae, etc. (Song et al., 2021; Liu et al., 2022).

REVISED.
Furthermore, phage preparations may also be useful as living disinfectants for decontaminating or cleaning surfaces, such as in hospitals. They are particularly effective at killing bacteria within biofilms since biofilms are often resistant to chemical disinfectants. To date, it has been shown that for some bacteria, such as Pseudomonas aeruginosa, chemical disinfectants like sodium hypochlorite and benzalkonium chloride have synergistic effects with some phages, enhancing biofilm elimination on contaminated surfaces. Similarly, phages that can eliminate biofilms of other important bacterial pathogens such as S. aureus, Escherichia coli, A. baumannii, K. pneumoniae, etc., have been reported (Song et al., 2021; Liu et al., 2022).

Experimental design

To the best of my knowledge the authors have made a comprehensive and unbiased coverage of the subject.

Validity of the findings

Although the authors present a well-organized and comprehensive survey of the appropriate literature, in my opinion, their conclusions are limited. I encourage the authors to be more speculative (within reasonable limits) regarding the limitations and future developments of the use of bacteriophages.

---

## Round 0.2 · accepted · Accept

Dear authors,

The previous Academic Editor is unavailable and so I am handling the manuscript in my capacity as Section Editor.

Thank you so much for submitting your work to PeerJ. Based on the previous reports by two experts in the field, the major concerns were the writing and grammar issues found and other issues that have been addressed in full. The new revised version of the manuscript is now suitable for publication. I thank the authors for addressing the issues and I thank the two experts that provided positive insight to this manuscript. Congratulations!

Best regards,

Kostas